

# Deep learning prediction of likelihood of ICU admission and mortality in COVID-19 patients using clinical variables

Xiaoran Li[1],[*], Peilin Ge[1],[*], Jocelyn Zhu[1], Haifang Li[1], James Graham[1], Adam Singer[2], Paul S. Richman[3] and Tim Q. Duong[4]

[1] Department of Radiology, Renaissance School of Medicine, Stony Brook University, New York, Stony Brook, NY, USA
[2] Department of Emergency Medicine, Renaissance School of Medicine, Stony Brook University, New York, Stony Brook, NY, USA
[3] Department of Medicine, Renaissance School of Medicine, Stony Brook University, New York, Stony Brook, NY, USA
[4] Department of Radiology, Albert Einstein College of Medicine, Bronx, NY, USA
[*] These authors contributed equally to this work.

Corresponding author
Tim Q. Duong,
tim.duong@einsteinmed.org

## ABSTRACT

**Background:** This study aimed to develop a deep-learning model and a risk-score system using clinical variables to predict intensive care unit (ICU) admission and in-hospital mortality in COVID-19 patients.

**Methods:** This retrospective study consisted of 5,766 persons-under-investigation for COVID-19 between 7 February 2020 and 4 May 2020. Demographics, chronic comorbidities, vital signs, symptoms and laboratory tests at admission were collected. A deep neural network model and a risk-score system were constructed to predict ICU admission and in-hospital mortality. Prediction performance used the receiver operating characteristic area under the curve (AUC).

**Results:** The top ICU predictors were procalcitonin, lactate dehydrogenase, C-reactive protein, ferritin and oxygen saturation. The top mortality predictors were age, lactate dehydrogenase, procalcitonin, cardiac troponin, C-reactive protein and oxygen saturation. Age and troponin were unique top predictors for mortality but not ICU admission. The deep-learning model predicted ICU admission and mortality with an AUC of 0.780 (95% CI [0.760–0.785]) and 0.844 (95% CI [0.839–0.848]), respectively. The corresponding risk scores yielded an AUC of 0.728 (95% CI [0.726–0.729]) and 0.848 (95% CI [0.847–0.849]), respectively.

**Conclusions:** Deep learning and the resultant risk score have the potential to provide frontline physicians with quantitative tools to stratify patients more effectively in time-sensitive and resource-constrained circumstances.

## INTRODUCTION

Since the first reports of severe respiratory illness caused by coronavirus disease 2019 (COVID-19) in Wuhan, China in mid-December 2019 (*Huang et al., 2020*; *Zhu et al., 2020*), over 6.2 million individuals have been infected, resulting in over 370,000 deaths

worldwide (31 May 2020). The actual numbers are likely to be much higher due to testing shortages and under-reporting (*Yelin et al., 2020*). Many patients have mild or asymptomatic infections, while others deteriorate rapidly with multi-organ failure. There will likely be recurrence and secondary waves of this pandemic (*Leung et al., 2020*).

A large array of clinical and demographic variables associated with COVID-19 infection have been identified (see reviews *Brown et al., 2020*; *Cao et al., 2020*; *Rodriguez-Morales et al., 2020*). A few of these have been associated with high likelihood of critical illness or mortality. There are however no established prognostic models that reliably predict the need for escalated (intensive care unit, ICU) care or mortality due to COVID-19 infection. Lacking this, effective triage of patients is challenging in a resource-constrained environment. The problem is further magnified by the poor sensitivity (*Kim, Hong & Yoon, 2020*) and a few day turnaround time (*Yelin et al., 2020*) of the most commonly used reverse-transcriptase polymerase chain reaction (RT-PCR) test, during which time patients are assumed COVID-19 positive. This problem strains the resources of many hospitals and highlights the need for effective tools to anticipate patients' progression and properly triage patients.

The goal of this study was to develop a deep-learning algorithm (in contrast to previous methods) to identify the top, statistically significant predictors amongst the large array of clinical variables at admission to predict the likelihood of ICU admission and in-hospital mortality in COVID-19 patients. We further developed a simplified risk-score model to predict the likelihood of ICU admission and in-hospital mortality.

## METHODS

### Study population

This retrospective study was approved by Institutional Review Board with exemption of informed consent and HIPAA waiver (IRB-2020-00207; Stony Brook University Hospital, Stony Brook, NY, USA). Stony Brook University Hospital, the only academic hospital serving Suffolk county, about 40 miles east of New York City, was one of the hardest hit counties in the country at the time of this writing. The COVID-19 Persons Under Investigation (PUI) registry consisted of 5,766 patients from 7 February 2020 to 4 May 2020. Only patients who were diagnosed by positive tests of real-time polymerase chain reaction (RT-PCR) for severe acute respiratory syndrome coronavirus 2 (SARS-CoV-2) were included in the study. Demographic information, chronic comorbidities, imaging findings, vital signs, symptoms, and laboratory tests at admission were collected. Imaging findings were extracted from patient chart review, which included information provided by radiology report as part of standard of care. The primary outcome was ICU admission versus general floor admission, and the secondary outcome was in-hospital mortality versus discharge. Mortality outside of hospital after discharge was not obtained.

Figure 1 shows the flowchart of patient selection. Of the 2,594 confirmed COVID-19 positive cases, all 1,108 hospitalized COVID-19 positive patients were used in our analysis. Seventy-seven (77) patients were admitted to the ICU directly and an additional 194 patients were subsequently upgraded to an ICU from a general floor. Among these 271 ICU patients, 108 were discharged alive, 77 expired during the hospitalization and the

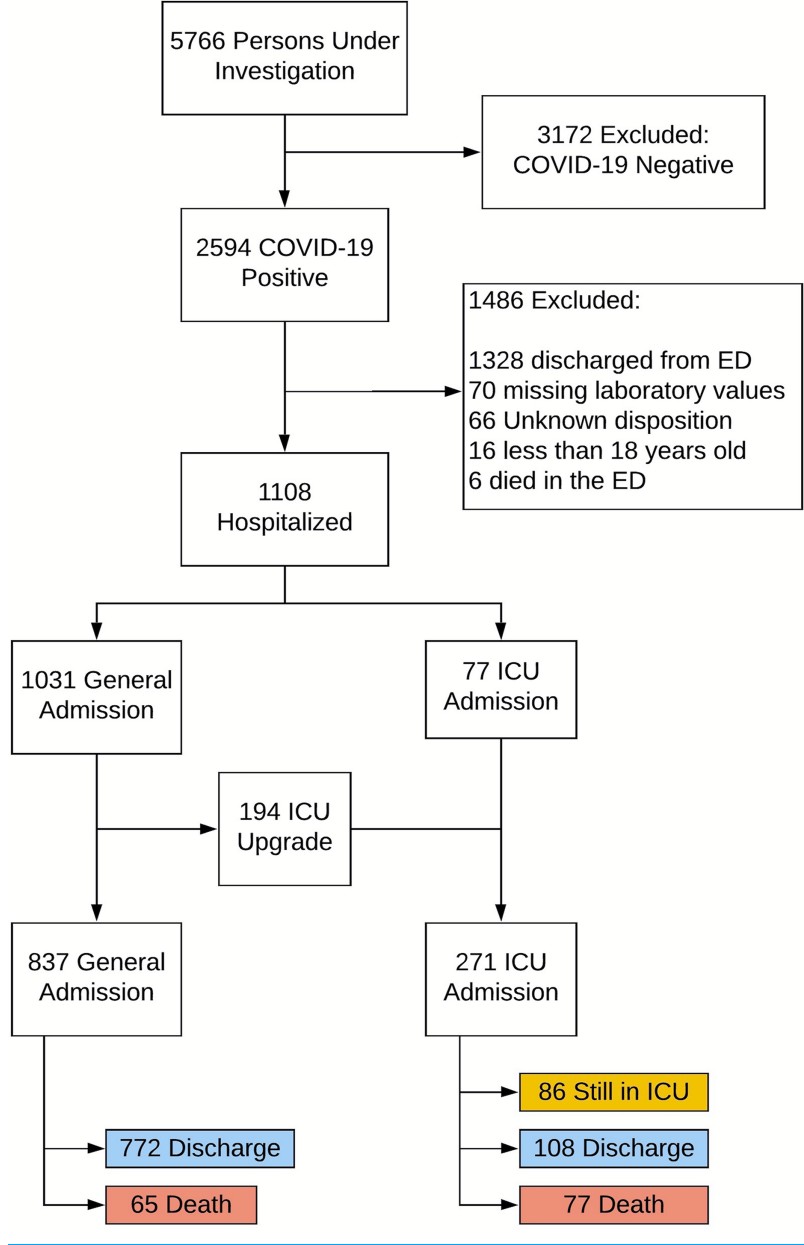

**Figure 1 Patient selection flowchart.** Patient selection flowchart.

other 86 are still in the hospital at the time of this analysis. Comparison was made to 837 general admissions who did not receive ICU care, among whom 772 patients were discharged alive and 65 expired during the hospitalization (none remained in the hospital).

## Data preprocessing

Two patients were excluded from machine learning analysis for missing categorical variables. Brain natriuretic peptide (BNP) was missing from >15% of patients, thus they were excluded from machine learning analysis. For the rest of the laboratory variables, missing data (in <5% of patients) was imputed with predictive mean modeling using the

Multivariate Imputation by Chained Equations in R (statistical analysis software, version 4.0) (*Van Buuren & Groothuis-Oudshoorn, 2011*).

## Deep neural network prediction model

Ranking of clinical variables of categorical or numerical values were made using the Boruta, a statistical software (*Kursa & Rudnicki, 2010*). Boruta ranks feature importance using the Random Forest method. In this decision tree-based method, the quantitative measure of importance is the Gini feature of importance, which counts the times that a feature is used to split a node of a decision tree, statistically weighted by the number of instances the node splits. In the DNN model, the top predictors were those that demonstrated statistical significance using built-in statistical methods within the Boruta algorithm.

A correlation coefficient >0.5 from collinearity analysis was used to exclude correlated variables from machine learning analysis. Note that none of the top features we used in the final analysis demonstrated strong correlation with other features. Thus, no top features were removed as a result. A deep neural network (DNN) was constructed to predict ICU admission and mortality using five fully connected dense layers (*Chen et al., 2020*). The top clinical predictors were input parameters, determined by testing subsets of these parameters, and ICU admission and mortality were outcome parameters. The DNN model used five hidden layers with 6, 8, 16, 8, 4 neurons respectively. We explored a few models using a range of number (3–7) of layers, and the 5-layer model yielded the optimal validation result. ReLu activation function for the hidden layers, the sigmoid activation function for the output layer, and the "he_normal" normalization scheme were applied. In the model training process, we used Adam optimizer, mean squared error as the cost function, a default learning rate of 0.01, and number of epochs of 100. The reported results yielded from the average of five consecutive runs. The dataset was randomly split into 90% training data and 10% testing data. ICU admission and mortality results were categorized using a binary classification. To minimize overfitting, we employed 5-fold cross-validation, ranked and removed less important features using correlation analysis and based on statistical significance by Boruta. We also employed regularization and stopped the training process at 100 epochs.

## Risk score model

Risk-score systems were constructed using the top independent clinical variables to predict ICU admission and mortality. For risk score, the mixed Generalized Additive Model was used to plot the probability of ICU admission and mortality for each clinical variable *Wood & Augustin (2002)*. Different cutoff points were evaluated where the chosen cutoff points yielded the optimal distribution (not skewed to high or low scores) of the risk score model. The corresponding numerical values of each top feature at probability of 0.3 for ICU and 0.2 for mortality were found to be the optimal cutoff values for the risk score model. Each of the top variables was assigned a weight of one point if the clinical measurement was above the probability cutoff. The risk score ranged from 0 to 5 for ICU admission and 0–6 for mortality (which were chosen based on statistical significance, see "Results").

## Statistical analysis and performance evaluation

Statistical analysis was performed in SPSS v26 and in R (statistical analysis software 4.0). Group comparisons of categorical variables in frequencies and percentages used the chi-square test or Fisher exact test. Group comparison of continuous variables in medians and interquartile ranges (IQR) used the Mann–Whitney $U$ test. A $p$ value $< 0.05$ was considered to be statistically significant. For performance evaluation, data were split 90% for training and 10% for testing. Prediction performance was evaluated by calculating the area under the curve (AUC) of the receiver operating characteristic (ROC) curve, accuracy, sensitivity, specificity, precision, recall, negative predictive value (NPV), positive predictive value (PPV) and F1 score (a harmonic mean of precision and recall). The average ROC analysis was repeated with five runs. In risk score models, SPSS was used to cross-check statistical significance of the top features, in which all top features used in the final analysis of risk score model had a $p < 0.001$.

# RESULTS

## Clinical variables associated with ICU admission

Table 1 summarizes the demographic characteristics, vital signs, comorbidities and laboratory data for the ICU ($n = 271$) and non-ICU ($n = 837$) group. The median age of the ICU group was lower than that of the general admission group (59 years (IQR: 49–71) vs. 62 years (IQR: 50–76), $p = 0.027$). Disproportionally more males were admitted to the ICU (67.5% vs. 32.5%, $p < 0.001$). History of cancer was the only comorbidity that was significantly associated with ICU admission ($P = 0.016$).

All measured vital signs were significantly different between the ICU group and the non-ICU group. The ICU group had higher heart rate, respiratory rate and temperature, but lower systolic blood pressure and oxygen saturation ($p < 0.05$). The ICU group had higher alanine aminotransferase (ALT), C-reactive protein (CRP), D-dimer, ferritin, lactate dehydrogenase (LDH), white blood cells (WBC) and procalcitonin ($p < 0.05$) and lower lymphocyte counts ($p < 0.05$). Cardiac troponin and BNP were not significantly different between groups ($p > 0.05$).

The symptom of dyspnea was significantly associated with ICU admission ($p = 0.001$). Patients admitted to ICU were more likely to present with abnormal chest x-ray ($p < 0.001$), and more likely to have bilateral chest x-ray abnormalities on presentation, compared to that of general admission group ($p < 0.001$).

## Prediction models for ICU admission

Figure 2 shows the ranking of the clinical variables associated with ICU admission. The top five statistically significant predictors of ICU admission were procalcitonin, LDH, CRP, ferritin, and SpO2. A deep neural network predictive model for mortality was constructed using the top clinical variables and trained using the training dataset and tested on an independent testing dataset. The ROC and confusion matrix of the testing dataset are shown in Fig. 3. The performance of the DNN model yielded an AUC = 0.780 (95% CI [0.760–0.785]), sensitivity = 0.760, specificity = 0.709 and F1 score = 0.551 in predicting ICU admission for the testing set (Table 2).

Table 1 **Demographic characteristics, comorbidities, symptoms, imaging findings, vital signs and laboratory findings of ICU versus non-ICU patients.** Demographic characteristics, comorbidities, symptoms, imaging findings, vital signs, and laboratory findings of ICU versus non-ICU patients. Group comparison of categorical variables in frequencies and percentages used $\chi^2$ test or Fisher exact tests. Group comparison of continuous variables in medians and interquartile ranges (IQR) used the Mann–Whitney $U$ test.

| | Patients, no. (%) | | |
| | ICU (n = 271) | Non-ICU (n = 837) | p Value |
|---|---|---|---|
| Demographics | | | |
| Age, median (range), year | 59 (49, 71) | 62 (50,76) | 0.027 |
| Sex | | | <0.001 |
| Male | 183 (67.5%) | 452 (54.0%) | |
| Female | 88 (32.5%) | 385 (46.0%) | |
| Ethnicity | | | 0.153 |
| Hispanic/latino | 78 (28.8%) | 223 (26.6%) | |
| Non-hispanic/latino | 148 (54.6%) | 507 (60.6%) | |
| Unknown | 45 (16.6%) | 107 (12.8%) | |
| Race | | | 0.003 |
| Caucasian | 123 (45.4%) | 453 (54.1%) | |
| African American | 13 (4.8%) | 61 (7.3%) | |
| Asian | 20 (7.4%) | 26 (3.1%) | |
| American Indian/Alaska Native | 2 (0.7%) | 2 (0.2%) | |
| Native Hawaiian or other Pacific Islander | 0 | 1 (0.1%) | |
| More than one race | 0 | 5 (0.6%) | |
| Unknown/not reported | 113 (41.7%) | 289 (34.5%) | |
| Comorbidities | | | |
| Smoking history | 61 (22.6%) | 214 (25.6%) | 0.332 |
| Diabetes | 80 (29.5%) | 220 (26.3%) | 0.308 |
| Hypertension | 126 (46.5%) | 412 (49.3%) | 0.442 |
| Asthma | 23 (8.5%) | 43 (5.1%) | 0.054 |
| COPD | 39 (14.4%) | 126 (15.1%) | 0.845 |
| Coronary artery disease | 17 (6.3%) | 76 (9.1%) | 0.166 |
| Heart failure | 18 (6.6%) | 62 (7.4%) | 0.787 |
| Cancer | 15 (5.5%) | 88 (10.5%) | **0.016** |
| Immunosuppression | 20 (7.4%) | 64 (7.7%) | 1.000 |
| Chronic kidney disease | 20 (7.4%) | 81 (9.7%) | 0.276 |
| Symptoms | | | |
| Fever | 191 (70.5%) | 547 (65.4%) | 0.138 |
| Cough | 191 (70.5%) | 564 (67.4%) | 0.368 |
| Shortness of breath | 210 (77.5%) | 557 (66.5%) | **0.001** |
| Fatigue | 56 (20.7%) | 201 (24.0%) | 0.282 |
| Sputum | 25 (9.2%) | 50 (6.0%) | 0.071 |
| Myalgia | 61 (22.5%) | 192 (22.9%) | 0.934 |
| Diarrhea | 60 (22.1%) | 201 (24.0%) | 0.565 |

| | Patients, no. (%) | | |
| --- | --- | --- | --- |
| | ICU (n = 271) | Non-ICU (n = 837) | p Value |
| Nausea or vomiting | 48 (17.7%) | 176 (21.0%) | 0.258 |
| Sore throat | 21 (7.7%) | 61 (7.3%) | 0.790 |
| Rhinorrhea | 14 (5.2%) | 36 (4.3%) | 0.613 |
| Loss of smell | 11 (4.1%) | 34 (4.1%) | 1.000 |
| Loss of taste | 12 (4.4%) | 42 (5.0%) | 0.871 |
| Headache | 80 (9.6%) | 28 (10.3%) | 0.724 |
| Chest discomfort or chest pain | 43 (15.9%) | 133 (15.9%) | 1.000 |
| Imaging studies | | | |
| Abnormal chest x-ray results | 227 (92.1%) | 694 (83.6%) | <0.001 |
| Chest x-ray findings | | | <0.001 |
| Unilateral | 26 (10.7%) | 140 (20.7%) | |
| Bilateral | 218 (89.3%) | 536 (79.3%) | |
| Vital signs, median (IQR) | | | |
| Heart Rate, bpm | 100 (87, 115) | 98 (83, 110) | 0.003 |
| Respiratory rate, rate/min | 23 (18, 30) | 20 (18, 24) | <0.001 |
| SpO$_2$ % | 93 (87, 96) | 94 (92, 97) | <0.001 |
| Systolic blood pressure, mmHg | 122 (108, 137) | 127 (114, 144) | 0.003 |
| Temperature, °C | 37.4 (36.9, 38.3) | 37.3 (36.9, 38.0) | 0.021 |
| Laboratory findings at admission, median (IQR) | | | |
| Alanine aminotransferase, U/L | 37 (22, 59) | 29 (17, 51) | <0.001 |
| Brain natriuretic peptide, pg/mL | 276 (81, 1123) | 212 (53, 1143) | 0.177 |
| C-reactive protein, mg/dL | 12.8 (6.9, 22.1) | 7.2 (3.2, 13.3) | <0.001 |
| D-dimer, ng/mL | 401 (257, 831) | 353 (217, 657) | 0.012 |
| Ferritin, ng/mL | 1132 (582, 1867) | 613 (289, 1234) | <0.001 |
| Lactate dehydrogenase, U/L | 436 (332, 593) | 332 (257, 433) | <0.001 |
| WBC, ×10$^3$/ml | 8.1 (6.1, 11.6) | 7.3 (5.5, 9.4) | 0.001 |
| Lymphocytes, % | 10.6 (6.1, 15.4) | 13.1 (8.4, 19.5) | <0.001 |
| Procalcitonin, ng/mL | 0.29 (0.16, 0.77) | 0.15 (0.09, 0.28) | <0.001 |
| Troponin, ng/mL | 0.01 (0.01, 0.01) | 0.01 (0.01, 0.01) | 0.596 |

**Note:**
COPD, chronic obstructive pulmonary disease; IQR, interquartile range; SpO$_2$, oxygen saturation. SI conversion factors: To convert alanine aminotransferase and lactate dehydrogenase to microkatal per liter, multiply by 0.0167; C-reactive protein to milligram per liter, multiply by 10; D-dimer to nanomole per liter, multiply by 0.0054; leukocytes to ×10$^9$ per liter, multiply by 0.001.

A risk score system was constructed (training data set) using the top five statistically significant clinical variables, with one point given for each variable meeting the following criteria: procalcitonin > 0.5 ng/mL, LDH >487 U/L and <12,586.7 U/L, CRP > 14.2 mg/dL, ferritin > 1,250 ng/mL and <13,080.5 ng/mL and SpO2 < 88.8%. Odds ratios of procalcitonin, LDH, CRP, ferritin and SpO2 for ICU admission were 3.062, 3.846, 3.001, 2.449 and 3.665, respectively. Figure 4 shows the results for the testing data set using the risk score system. ICU admission rate increased with increasing risk scores.

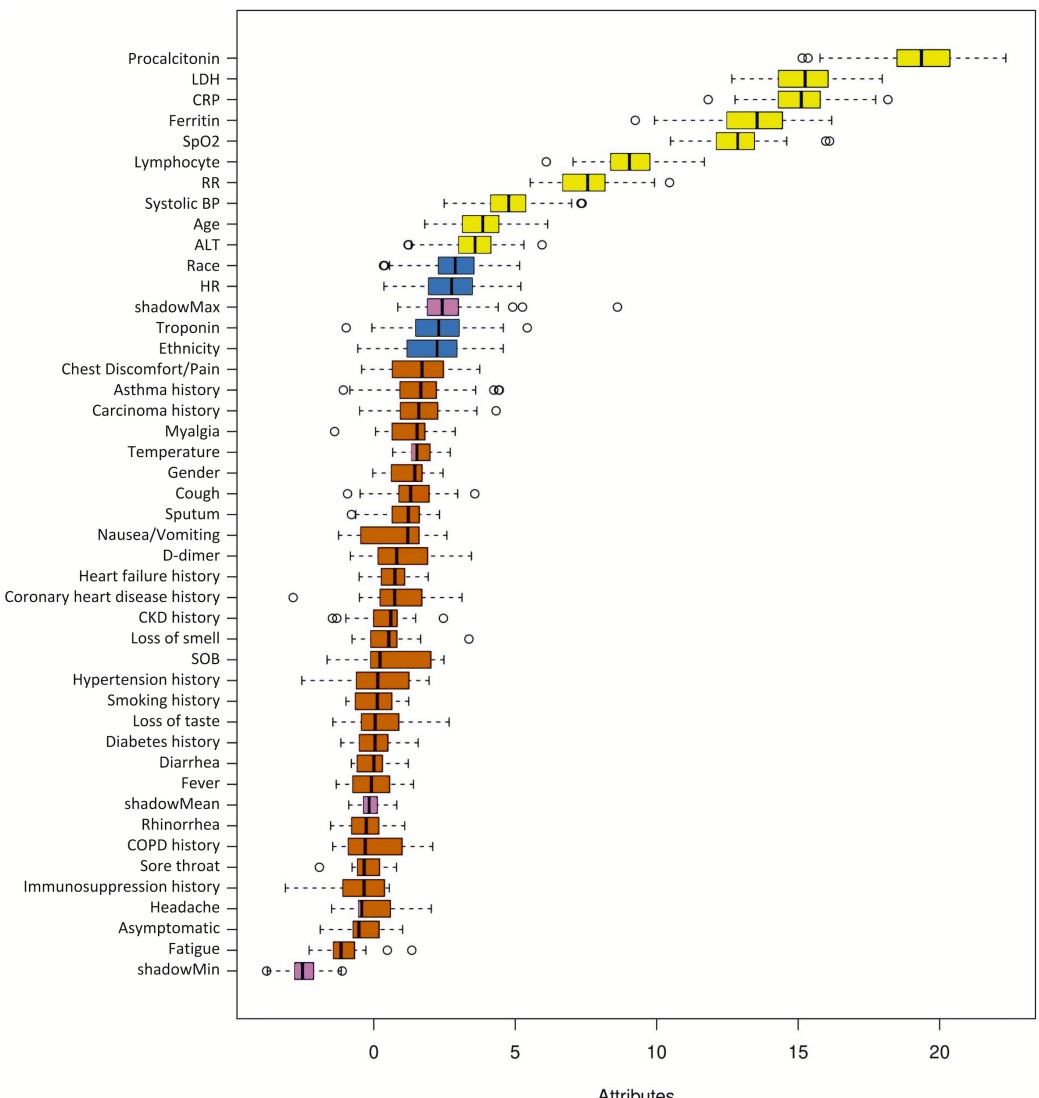

**Figure 2 Ranking of clinical variables for predicting ICU admission.** Ranking of clinical variables for predicting ICU admission by Boruta algorithm. The *x*-axis is attribute of level of importance, where a larger number indicates relatively higher importance. The *y*-axis are laboratory test variables. The top statistically significant predictors were: procalcitonin, LDH, CRP, ferritin, SpO2, lymphocytes, respiratory rate, systolic blood pressure, age and ALT. The top 10 variables were significant.

The performance of the risk score yielded an AUC of 0.728 (95% CI [0.726–0.729]) for predicting ICU admission for the testing data set.

## Clinical variables associated with mortality

Table 3 summarizes the demographic data, vital signs, comorbidities and laboratory data for the non-survivors (*n* = 142) and survivors (*n* = 880) group. The median age of the non-survivor group was higher than that of the survivor group (76 years (IQR: 66–84) vs. 59 years (IQR: 49–72), *p* < 0.001). There was a disproportionally higher mortality rate

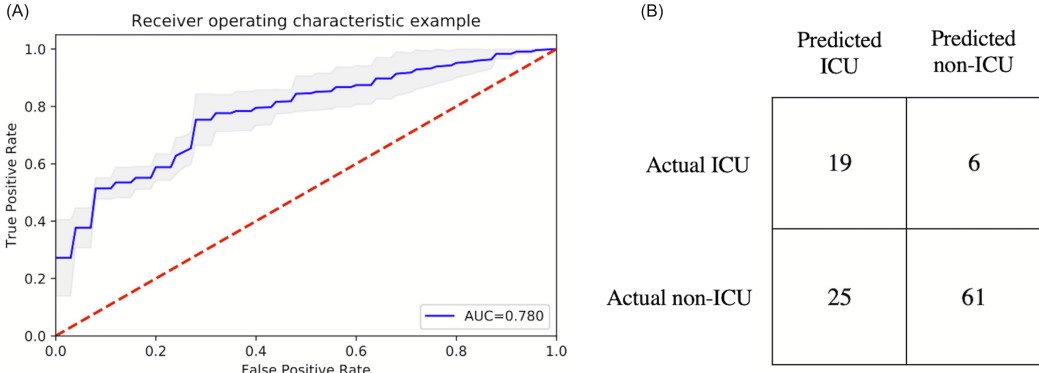

**Figure 3 ROC and confusion matrix for prediction of ICU admission.** (A) ROC and (B) confusion matrix for prediction of ICU admission of the DNN model.

**Table 2 Performance indices for predicting ICU admission of the testing dataset.** Performance indices for predicting ICU admission of the testing dataset. Abbreviations: area under the curve (AUC), accuracy, sensitivity, specificity, precision, recall, negative predictive value (NPV), positive predictive value (PPV) and F1 score (a harmonic mean of precision and recall).

|  | AUC | Accuracy | Sensitivity | Specificity | Precision | NPV | PPV | F1 |
|---|---|---|---|---|---|---|---|---|
| Training | 0.751 | 0.703 | 0.707 | 0.701 | 0.437 | 0.879 | 0.437 | 0.540 |
| Testing | 0.728 | 0.721 | 0.760 | 0.709 | 0.432 | 0.910 | 0.432 | 0.551 |

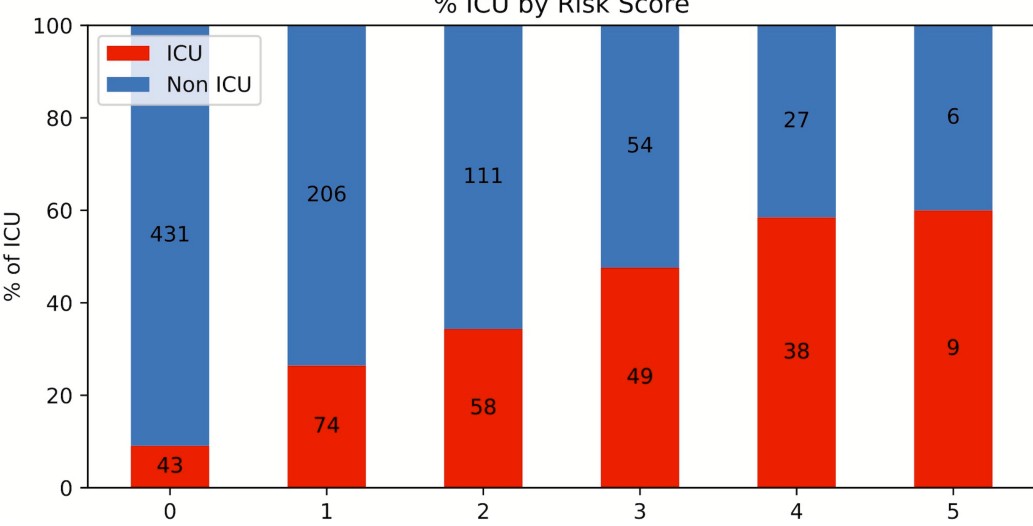

**Figure 4 Risk score stratification for ICU admission.** Risk score stratification for ICU admission. Scores ranged from 0 to 5, with 0 indicating the lowest risk and 5 being the highest risk of mortality. The numbers in the bar indicate the number of patients in the ICU (red) and non-ICU (blue) that were correctly predicted in the testing dataset.

in males (65.5% vs. 34.5%, $p = 0.014$). Of the comorbidities, hypertension, coronary artery disease, heart failure, chronic obstructive pulmonary disease, smoking history and chronic kidney disease were significantly different between groups ($p < 0.05$).

**Table 3 Demographic characteristics, comorbidities, symptoms, imaging findings, vital signs, and laboratory findings of death versus non-death (discharged).** Demographic characteristics, comorbidities, symptoms, imaging findings, vital signs, and laboratory findings of death versus non-death (discharged). Group comparison of categorical variables in frequencies and percentages used $\chi^2$ or Fisher exact tests. Group comparison of continuous variables in medians and interquartile ranges (IQR) used the Mann–Whitney $U$ test.

| | Patients, no. (%) | | |
| --- | --- | --- | --- |
| | Death (n = 142) | Non-death (n = 880) | p Value |
| Demographics | | | |
| Age, median (range), year | 76 (66,84) | 59 (49,72) | <0.001 |
| Sex | | | 0.022 |
| Male | 93 (65.5%) | 484 (55.0%) | |
| Female | 49 (34.5%) | 396 (45.0%) | |
| Ethnicity | | | 0.001 |
| Hispanic/Latino | 23 (16.2%) | 251 (28.5%) | |
| Non-Hispanic/Latino | 105 (73.9%) | 504 (57.3%) | |
| Unknown | 14 (9.9%) | 125 (14.2%) | |
| Race | | | 0.023 |
| Caucasian | 91 (64.1%) | 450 (51.1%) | |
| African American | 6 (4.2%) | 61 (6.9%) | |
| Asian | 9 (6.3%) | 33 (3.8%) | |
| American Indian/Alaska Native | 1 (0.7%) | 2 (0.2%) | |
| Native Hawaiian or other Pacific Islander | 0 | 1 (0.1%) | |
| More than one race | 0 | 5 (0.6%) | |
| Unknown/not reported | 35 (24.6%) | 328 (37.3%) | |
| Comorbidities | | | |
| Smoking history | 52 (36.6%) | 204 (23.2%) | 0.001 |
| Diabetes | 48 (33.8%) | 229 (26.1%) | 0.067 |
| Hypertension | 92 (64.8%) | 402 (45.8%) | <0.001 |
| Asthma | 6 (4.2%) | 51 (5.8%) | 0.557 |
| COPD | 23 (16.2%) | 66 (7.5%) | 0.002 |
| Coronary artery disease | 39 (27.5%) | 115 (13.1%) | <0.001 |
| Heart failure | 29 (20.4%) | 47 (5.4%) | <0.001 |
| Cancer | 19 (13.4%) | 78 (8.9%) | 0.092 |
| Immunosuppression | 8 (5.6%) | 65 (7.4%) | 0.598 |
| Chronic kidney disease | 20 (14.1%) | 75 (8.5%) | 0.043 |
| Symptoms | | | |
| Fever | 81 (57.0%) | 599 (68.1%) | 0.012 |
| Cough | 73 (51.4%) | 628 (71.4%) | <0.001 |
| Shortness of breath | 102 (71.8%) | 594 (67.5%) | 0.333 |
| Fatigue | 19 (13.4%) | 216 (24.5%) | 0.003 |
| Sputum | 10 (7.0%) | 58 (6.6%) | 0.856 |
| Myalgia | 15 (10.6%) | 220 (25.0%) | <0.001 |
| Diarrhea | 27 (19.0%) | 211 (24.0%) | 0.239 |

| | Patients, no. (%) | | |
| --- | --- | --- | --- |
| | Death (n = 142) | Non-death (n = 880) | p Value |
| Nausea or vomiting | 10 (7.0%) | 192 (21.8%) | <0.001 |
| Sore throat | 7 (4.9%) | 69 (7.8%) | 0.300 |
| Rhinorrhea | 4 (2.8%) | 41 (4.7%) | 0.386 |
| Loss of smell | 2 (1.4%) | 38 (4.3%) | 0.106 |
| Loss of taste | 2 (1.4%) | 48 (5.5%) | 0.035 |
| Headache | 7 (4.9%) | 90 (10.2%) | 0.045 |
| Chest discomfort or chest pain | 10 (7.0%) | 151 (17.2%) | 0.001 |
| Imaging studies | | | |
| Abnormal chest x-ray results | 123 (87.2%) | 720 (84.6%) | 0.524 |
| Chest x-ray findings | | | 0.214 |
| Unilateral | 18 (14.6%) | 142 (19.7%) | |
| Bilateral | 105 (85.4%) | 577 (80.3%) | |
| Vital signs, median (IQR) | | | |
| Heart Rate, bpm | 96 (81, 115) | 99 (85, 110) | 0.496 |
| Respiratory rate, rate/min | 24 (20, 32) | 20 (18, 24) | <0.001 |
| $SpO_2$ % | 93 (88, 96) | 94 (92, 96) | <0.001 |
| Systolic blood pressure, mmHg | 127 (105, 142) | 125 (113, 143) | 0.568 |
| Temperature, °C | 37.1 (36.7, 37.6) | 37.3 (36.9, 38.1) | <0.001 |
| Laboratory findings at admission, median (IQR) | | | |
| Alanine aminotransferase, U/L | 30.0 (17.0, 54.0) | 30.0 (18.0, 52.0) | 0.666 |
| Brain natriuretic peptide, pg/mL | 1652 (452, 4556) | 164 (47, 772) | <0.001 |
| C-reactive protein, mg/dL | 13.4 (6.9, 21.8) | 7.5 (3.2, 13.4) | <0.001 |
| D-dimer, ng/mL | 635 (365, 1753) | 333 (213, 606) | <0.001 |
| Ferritin, ng/mL | 981 (442, 1657) | 640 (308, 1333) | <0.001 |
| Lactate dehydrogenase, U/L | 436 (330, 638) | 333 (257, 434) | <0.001 |
| WBC, $\times 10^3$/ml | 8.7 (6.4, 12.3) | 7.3 (5.5, 9.5) | 0.001 |
| Lymphocytes, % | 8.9 (5.3, 13.6) | 13.3 (8.7, 19.4) | <0.001 |
| Procalcitonin, ng/mL | 0.34 (0.18, 1.26) | 0.15 (0.090, 0.28) | <0.001 |
| Troponin, ng/mL | 0.02 (0.01, 0.07) | 0.01 (0.01, 0.01) | <0.001 |

**Note:**

COPD, chronic obstructive pulmonary disease; IQR, interquartile range; $SpO_2$, oxygen saturation. SI conversion factors: To convert alanine aminotransferase and lactate dehydrogenase to microkatal per liter, multiply by 0.0167; C-reactive protein to milligram per liter, multiply by 10; D-dimer to nanomole per liter, multiply by 0.0054; leukocytes to $\times 10^9$ per liter, multiply by 0.001.

Among vital signs, tachypnea and hypoxemia were significantly different between groups at presentation ($p < 0.05$). The expired cohort had higher BNP, CRP, D-dimer, ferritin, LDH, WBC, procalcitonin and cardiac troponin but lower lymphocytes ($p < 0.05$). ALT was not significantly different between groups.

Among the symptoms, cough, myalgia, nausea or vomiting, chest discomfort, fatigue, fever, loss of taste and headache were significantly different between groups ($p < 0.05$). There was no significant difference in x-ray findings between groups at presentation.

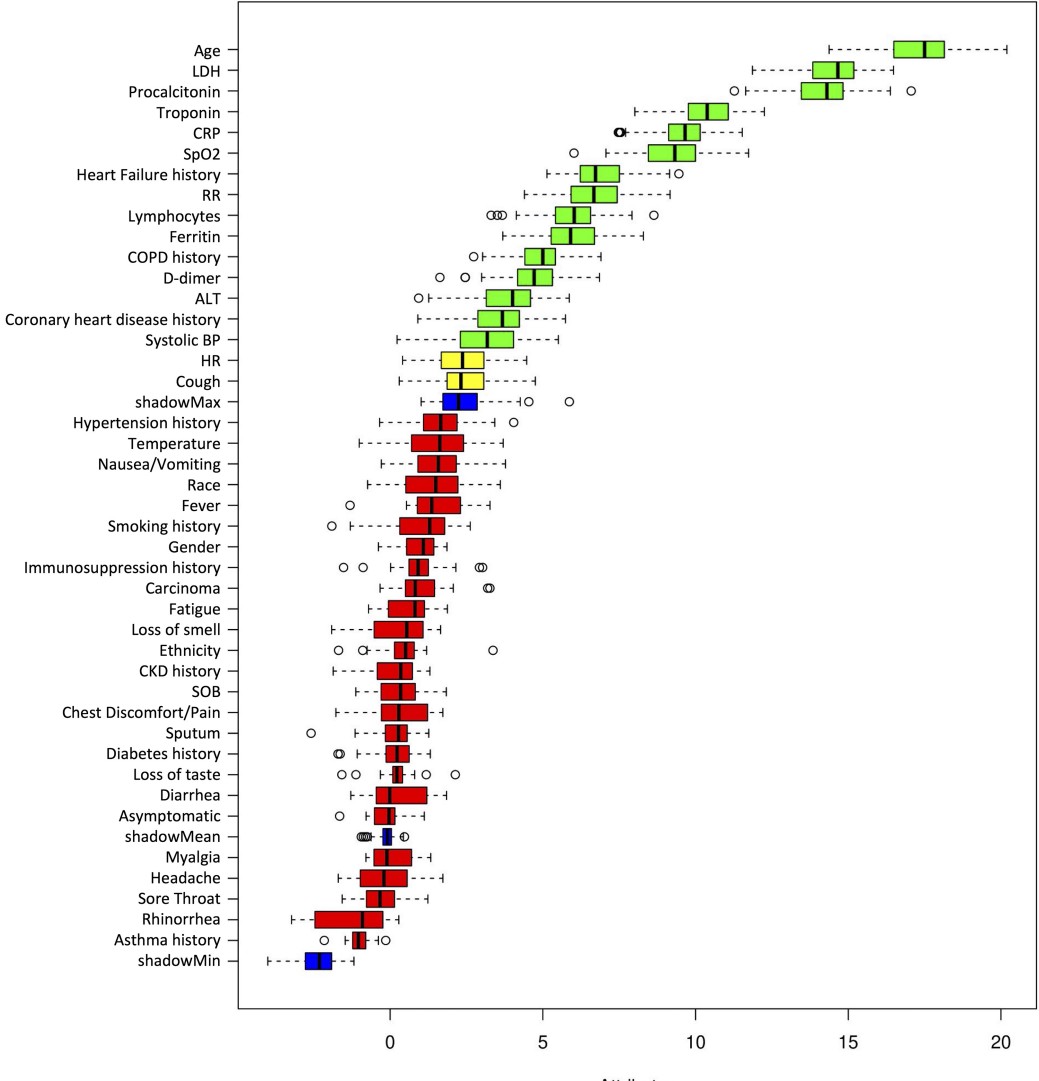

**Figure 5 Ranking of clinical variables for predicting mortality.** Ranking of clinical variables for predicting mortality by Boruta algorithm. The *x*-axis is attribute of level of importance, where a larger number indicates relatively higher importance. The *y*-axis are laboratory test variables. The top statistically significant predictors were: age, LDH, procalcitonin, troponin, CRP, SpO2, history of heart failure, respiratory rate, lymphocytes, ferritin, history of COPD, D-dimer, ALT, history of coronary heart disease and systolic blood pressure. The top 15 variables were significant.

## Prediction models for mortality

The top six statistically significant predictors of mortality were age, LDH, procalcitonin, troponin, CRP and SpO2 (Fig. 5). A deep neural network predictive model for mortality was constructed using the top clinical variables and trained using the training data set. The ROC and confusion matrix are shown in Fig. 6. The performance of the DNN model yielded an AUC of 0.844 (95% CI [0.839–0.848]), sensitivity = 0.750, specificity = 0.872 and F1 score = 0.616 for the testing dataset (Table 4).

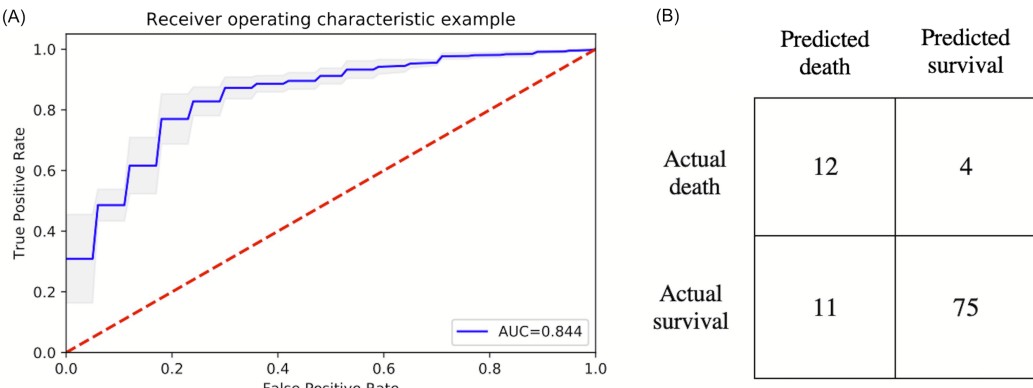

**Figure 6 ROC and confusion matrix for prediction of mortality.** (A) ROC and (B) confusion matrix for prediction of mortality of the DNN model.

**Table 4 Performance indices for predicting mortality.** Performance indices for predicting mortality of the testing dataset. Abbreviations: area under the curve (AUC), accuracy, sensitivity, specificity, precision, recall, negative predictive value (NPV), positive predictive value (PPV) and F1 score (a harmonic mean of precision and recall).

|  | AUC | Accuracy | Sensitivity | Specificity | Precision | NPV | PPV | F1 |
|---|---|---|---|---|---|---|---|---|
| Training | 0.852 | 0.892 | 0.706 | 0.922 | 0.589 | 0.952 | 0.589 | 0.642 |
| Testing | 0.844 | 0.853 | 0.750 | 0.872 | 0.522 | 0.949 | 0.522 | 0.616 |

A risk score system was constructed (training data set) using the top six statistically significant clinical variables to predict mortality. The thresholds for the risk scores were: age >71 years, LDH > 487 U/L, procalcitonin > 1.1 ng/mL, troponin > 0.03 ng/mL, CRP > 17 mg/dL and SpO2 < 88%. Odds ratios of age, LDH, procalcitonin, troponin, CRP and SpO2 for mortality were 4.301, 3.418, 6.232, 5.253, 4.240 and 3.750, respectively. Higher mortality rate was associated with higher risk scores for the testing set (Fig. 7). The performance of the risk score yielded an AUC of 0.848 (95% CI [0.847–0.849]) in predicting mortality for the testing set.

## DISCUSSION

Mining a large cohort of COVID-19 patients in the United States, deep-learning and resultant risk score models identified the top predictors of ICU admission in COVID-19 to be the admission levels of procalcitonin, LDH, CRP, ferritin and SpO2; the top predictors of mortality were age, LDH, procalcitonin, cardiac troponin, CRP and SpO2. Predictive models were developed using deep neural network of the top predictors, yielding an AUC of 0.779 and 0.882 for predicting ICU admission and mortality, respectively. The corresponding simplified risk scores yielded an AUC of 0.728 and 0.848, respectively.

The association between these biomarkers and poor outcomes in COVID-19 victims is biologically plausible: procalcitonin is elevated during bacterial infection, but less so during viral infection, suggesting that bacterial co-infection leads to worse outcome in COVID-19 patients (*Assicot et al., 1993*). LDH reflects tissue damage (*Huang et al., 2020*;

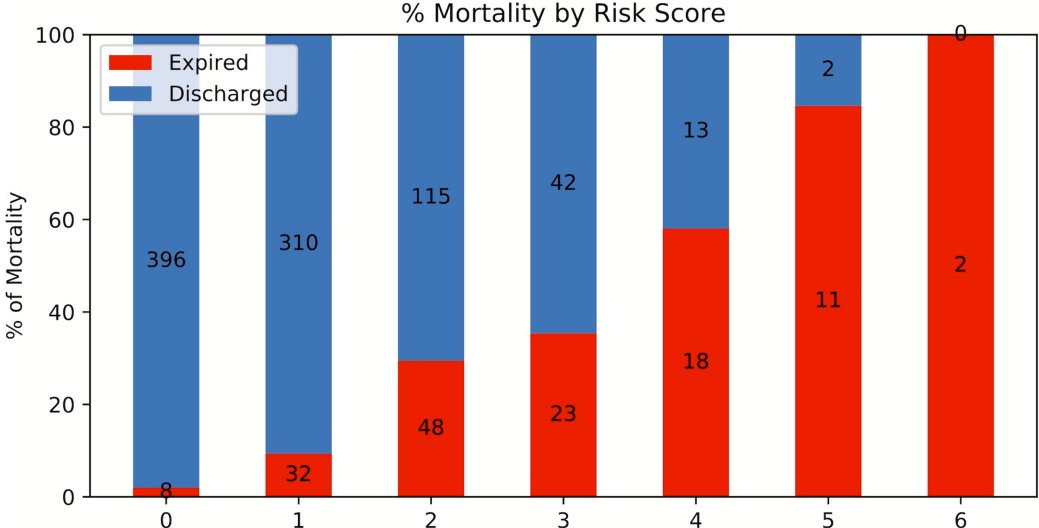

**Figure 7 Risk score stratification for mortality.** Risk score stratification for mortality. Scores ranged from 0 to 6, with 0 indicating the lowest risk and 6 being the highest risk of mortality. The numbers in the bar indicate the number of patients in the ICU (red) and non-ICU (blue) that were correctly predicted in the testing dataset.                           

*Zhu et al., 2020*), while CRP is indicative of inflammation (*Gabay & Kushner, 1999*). Elevated ferritin is associated with acute respiratory distress syndrome (ARDS) (*Connelly et al., 1997*) and may be a marker of aberrant iron metabolism that could render the lungs susceptible to oxidative damage (*Mumby et al., 2004*). Ferritin may reflect hyperinflammation associated with a cytokine storm and multi-organ failure (*Mehta et al., 2020*). Low SpO2 indicates failure of the lungs to oxygenate blood effectively, leading to tissue hypoxia (*Connelly et al., 1997*). Elevated cardiac troponin indicates cardiac injury (*Huang et al., 2020*). Although these variables have been previously associated with COVID-19 infection, most previous studies did not rank these clinical variables, or develop predictive models or risk scores to predict ICU admission or mortality. Not surprisingly, some of the same biomarkers in our study predicted both the need for ICU admission and likelihood of mortality. However, age and admission troponin level were uniquely predictive of mortality, indicating older age and cardiac issues are associated with higher rate of mortality in COVID-19 infection.

It is notable that individual comorbidities did not rank high in predicting ICU admission and mortality. Specifically, a history of heart failure, COPD, and coronary artery disease only ranked 7th, 11th and 14th respectively for predicting mortality. Similarly, the patients' symptoms and vital signs (other than SpO2) at the time of admission were not found to be the top predictors of poor outcome. Although some comorbidities have been reported to be associated with critical illness and mortality, most previously studies did not rank their importance with respect to other laboratory variables.

Our predictive AUC performance for ICU admission was poorer than that for mortality. We speculate this might be due to variability in triage decision-making to send patients to ICU among frontline clinicians. For both predictions, precision, PPV and F1 scores

were comparatively low, which was not unexpected due to the imbalanced sample sizes between the two groups as well as small sample sizes. Further studies are warranted.

While a large number of studies have previously identified clinical variables associated with the severity of COVID-19 infection, only a few studies have attempted to develop a predictive or risk score model to predict mortality and disease severity. *Jiang et al. (2020)* used supervised learning (not deep learning) and found mildly elevated alanine aminotransferase, myalgias and hemoglobin at presentation to be predictive of severe ARDS of COVID-19 with 70–80% accuracy. This study had small, non-uniform, heterogeneous clinical variables, obtained from different hospitals (*Jiang et al., 2020*). *Ji et al. (2020)* used logistic regression to predict stable versus progressive COVID-19 patients ($n = 208$) based on whether their conditions worsened during hospitalization. They reported comorbidities, older age, lower lymphocyte and higher lactate dehydrogenase at presentation to be independent high-risk factors for COVID-19 progression but did not develop a risk score. A nomogram of these four factors yielded a concordance index of 0.86. *Yan et al. (2020)* utilized supervised machine learning to predict critical COVID-19 at admission using presence of X-ray abnormality, cancer history, age, neutrophil/ lymphocyte ratio, LDH, dyspnea, bilirubin, unconsciousness and number of comorbidities. They reported an AUC of 0.88. *Yuan et al. (2020)* went one step further to predict mortality more than 12 days in advance with >90% accuracy across all cohorts. Moreover, their Kaplan–Meier score shows that patients upon admission could clearly be differentiated into low, medium or high risk. They created a simple risk score system, and validated using multiple independent cohorts (*Yuan et al., 2020*).

Our approach used a deep-learning algorithm which is novel and has distinct advantages over logistic regression and supervised learning approach. Deep learning is increasingly being used in medicine (*Deo, 2015*; *Santos et al., 2019*; *Tschandl et al., 2019*). In contrast to conventional analysis methods, which specify the relationships amongst data elements to outcomes, machine learning employs computer algorithms to identify relationships amongst different data elements to inform outcomes without the need to specify such relationships a priori. Deep learning can outperform human experts in performing many tasks in medicine (*Killock, 2020*). In addition to approximating physician skills, Deep learning can also detect novel relationships not readily apparent to human perception, especially in large, complex, and longitudinal datasets. Disadvantages of deep learning methods are that it requires comparatively large sample size, there is a potential of overfitting, and the complex relations could make deep learning results difficult to interpret, amongst others. In addition, we devised a simplified practical risk score adds practical utility to these findings. Although we ranked all variables and explicitly listed 10 or 15 top variables, we built the predictive model and risk score model using only the top five variables to simplify and increase translation potential in the clinical settings. The excellent prediction performances using a few clinical variables are encouraging.

This study has several limitations in addition to those mentioned above. This is a retrospective study carried out in a single hospital. These findings need to be replicated in large and multi-institutional settings for generalizability. We only analyzed clinical variables at admission. Longitudinal changes of these clinical variables need to be studied.

As in all observational studies, other residual confounders may exist that were not accounted for in our analysis. Future prospective studies validating our predictive models and scores are warranted.

## CONCLUSION

We implemented a deep-learning algorithm and a risk score model to predict the likelihood of ICU admission and mortality in COVID-19 patients. Our predictive model and risk score model can be easily retrained with additional data, new local data, as well as additional clinical variables. This approach has the potential to provide frontline physicians with a simple and objective tool to stratify patients based on risks so that COVID-19 patients can be triaged more effectively in time-sensitive, stressful and potentially resource-constrained environments.

## ACKNOWLEDGEMENTS

We thank all healthcare professionals for their hard work being at the front line of the pandemic.

### Funding
The authors received no funding for this work.

### Competing Interests
The authors declare that they have no competing interests.

### Author Contributions

- Xiaoran Li conceived and designed the experiments, performed the experiments, analyzed the data, prepared figures and/or tables, authored or reviewed drafts of the paper, and approved the final draft.
- Peilin Ge conceived and designed the experiments, performed the experiments, analyzed the data, prepared figures and/or tables, authored or reviewed drafts of the paper, and approved the final draft.
- Jocelyn Zhu performed the experiments, analyzed the data, prepared figures and/or tables, and approved the final draft.
- Haifang Li performed the experiments, analyzed the data, authored or reviewed drafts of the paper, and approved the final draft.
- James Graham performed the experiments, analyzed the data, prepared figures and/or tables, and approved the final draft.
- Adam Singer performed the experiments, analyzed the data, authored or reviewed drafts of the paper, and approved the final draft.
- Paul S. Richman performed the experiments, analyzed the data, authored or reviewed drafts of the paper, and approved the final draft.
- Tim Q. Duong analyzed the data, authored or reviewed drafts of the paper, and approved the final draft.

## Human Ethics

The following information was supplied relating to ethical approvals (i.e., approving body and any reference numbers):

Stony Brook University IRB approved this study (IRB2020-00207).

## Data Availability

Raw data is available in the Supplemental Files.

## Supplemental Information

Supplemental information for this article can be found online at http://dx.doi.org/10.7717/peerj.10337#supplemental-information.

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
