# Peer review of "Deep learning prediction of likelihood of ICU admission and mortality in COVID-19 patients using clinical variables"

_PeerJ, doi:10.7717/peerj.10337_

## Round 0.1 · original submission · Major Revisions

Thank you for allowing us to consider your manuscript which is handled by me (Tuan Nguyen). You manuscript has now been reviewed by 3 experts, and their comments are attached for your perusal. As you will see from their comments, although the reviewers recognize the relevance of your work, they raise some methodological issues that I would like to invite you to comment on.

As Academic Editor, I have also read you manuscript with great interest. I thought that the work is quite clinically relevant, but it should clarify a number of points as follows:

1. The ranking of clinical variables was done using a statistical software. Could you explain (preferably in simple language) what were the criteria for ranking. Was it based on some kind of statistical significance, variance explained, or the probability of being in a model? This needs more detailed explanation.

2. How do you define 'statistical significance'. In the presence of multiple tests of association, the 0.05 significance level is perhaps not a good threshold.

3. I am not clear on the way you remove both variables. You state that a correlation coefficient >0.5 from collinearity analysis was used to exclude correlated variables from machine learning analysis. Why the threshold of 0.5 was chosen?

4. How did you define mortality? 30-day in hospital? What about mortality outside hospital?

5. Could you distinguish between surgical or medical ICU admission?

6. I am a bit confused of terminologies. Did you use machine learning or Generalized Additive Model? It seems to me that you used the latter, and it should be stated so in the manuscript.

7. For transparency, I suggest that you provide the actual odds ratio for each variable (with the outcome being ICU admission and mortality).

Reviewer 1 ·

Basic reporting

See below

Experimental design

See below

Validity of the findings

See below

Additional comments

This retrospective study consisted of 5766 persons-under-investigation for
COVID-19 between February 7, 2020, and May 4, 2020. Demographics, chronic
comorbidities, vital signs, symptoms, and laboratory tests at admission were collected. A
deep neural network model and a risk score model were constructed to predict ICU
admission and mortality. Prediction performance used area under the curve (AUC) of the
receiver operating characteristic analysis (ROC). The top ICU predictors were
procalcitonin, lactate dehydrogenase, C-reactive protein, ferritin, and SpO2. The top
mortality predictors were age, lactate dehydrogenase, procalcitonin, cardiac troponin, C reactive protein, and SpO2.

The paper is well-written and contains interesting results. Minor comments include:
1) it would be good to make the code available for readers to reproduce the analysis;
2) Method: A deep neural network (DNN) was constructed to predict ICU admission and
mortality using five fully connected dense layers classification. Could the authors elaborate on why you choose the structure with 5 layers? How many neurons are there in each layer?
3) It would be essential to compare/discuss the findings with other using standard machine learning methods, such as (but not limited to) https://www.researchsquare.com/article/rs-41151/v1.

Reviewer 2 ·

Basic reporting

The paper proposes a method for utilising deep neural nets (DNNs) in predicting ICU admission likelihood for COVID-19 patients based on critical parameters. Despite the task being interesting, there are some major concerns:-

1. The ethical/research compliance approval pertains to "AI of lung images of COVID-19". The authors should clarify whether their study utilises lung images, or some numeric datasets (as the attached files in the raw data are .CSV and seem to pertain to numeric/categorical critical parameters). Please clarify this in the paper whether this data is derived based on the lung images or otherwise.
2. The title of the paper is ambiguous. "Deep-learning artificial intelligence prediction...". It is a well known factor that deep learning is a type of AI technique, so either use AI or deep learning in the title. Using both does not make clear sense for the context.
3. It would be better to have the figures in-text instead of appendices, which makes it difficult for the average reader of this journal.
4. The authors mention that the top 10 variables were significant in predicting ICU admission. Has the significance been medically correlated/verified or is it based on the experimental analysis by the authors?
5. There are several technical details from the point of view of AI missing in the paper- especially in terms of the DNN architecture utilised and its hyper-parameters, making the paper lack robustness.

Experimental design

The paper proposes utilisation of a deep neural network (DNN) for predicting ICU admission and mortality. However, their is insufficient details on the network choice and other hyperparameters:-

1. What was the basis to utilise 5 hidden layers for the DNN? Were smaller network architectures utilised?
2. The experiments use a 90% training dataset and 10% testing dataset for the study. With such a large amount of training data and experiments being only carried on the smaller testing data (10%) - were more conventional train test splits used e.g. 70-30%, 80-20% etc.? If not, please provide a reference on the choice of train-test split or more details on this choice, as it is a vital characteristic in AI experimentation.
3. It is mentioned that correlated variables were excluded from analysis (with corr. coff > 0.5), however, generally for DNNs, feature selection and dimensionality reduction are automatic and manual feature engineering is not required? Please justify the basis for identifying top predictors through this technique. There can be other methods e.g. XGBoost which can provide top predictors during classification, so a justification of choosing this methodology will help make this paper more robust.
4. Many relevant details are missing for the DNN which affect reproducibility and transparency in the results- what was the optimiser utilised, number of epochs over which the network was trained, activation function, learning rate.....?

Validity of the findings

1. The approach presents an evaluation of the DNN on the test dataset for COVID-19 patients in the US with an AUC. The confusion which is present is regarding utilisation of simpler ML architectures or DNNs with fewer hidden layers- were these verified and experimented with? If so, please provide more information in the text.

2. What were the key performance metrics alongside the AUC? True/False Positives/Negatives, F Score obtained for classification using the DNN, accuracy/precision/recall et al.? A confusion matrix here would be useful to clearly make inferences from the model's performance.

Additional comments

The paper proposes an interesting application of DNNs for a real-world (and very significant problem) at the present time of COVID-19. However, there are insufficient technical details on the AI part of this work, making replication/reproducibility a concern. In addition, the results in the paper need more details and justifications to merit publication in this journal.

Reviewer 3 ·

Basic reporting

The authors' writing is clear and easy to follow. However, there are some points could be improved:

1. The AUCs of models for mortality should be consistent in the abstract (0.848 and 0.845) and the text (0.882 and 0.048).
2. Authors mentioned that a deep-learning artificial intelligence algorithm is novel and has distinct advantages over logistic regression and supervised learning approach (line 237-238, 72-73) but did not give in any details of what advantages they are.
3. Figure 2 &4: More information should be provided: What values shown in the x-axis, which method used to retrieve these values, what information indicated by different colour codes?
4. Figure 3: Legend indicates that scores ranged from 0 to 6, while the figure represents score range from 0 to 5. Which information do the numbers in each bar indicate?
5. Table 1: Race values should be aligned. There are two bold p values without any indication. Lack of explanation for the WBC abbreviation.
6. Table 2: Race values should be aligned. Lack of explanation for the WBC abbreviation.
7. Line 145: p values should be included.
8. Line 209: lack of references.
9. Minor typo error: line 223: identified f clinical.

Experimental design

The research question is well-fined. The study settings is also clearly described. However, details of methods used in the study need to be provided:

10. Authors should explain in the text why Figure 2 mentions that top 10 variables were significant, but the models used only top 5 variables. Similarly, top 15 significant variables were mentioned in Figure 4 while the models used only top 6. If they were results from the collinearity analysis, please give details.

11. What did authors do to avoid the overfitting issue?

12. Although authors cited several references, all algorithms and parameter tuning strategies used in the study should be briefly described in the text or supplemental document to be useful for future reader. There is lack of references for the collinearity analysis.

Validity of the findings

Although your results are compelling, the data analysis should be improved in the following ways:
12. Provide ROC curves for training and test data sets.
13. Report performance of the models using other metrics, including sensitivity, specificity, accuracy, precision, and negative predictive value at the optional cut-off selected in training dataset because “change in AUC has little direct clinical meaning for clinicians” (Halligan et al, Eur Radiol, 2015, V25, p 932–939). These values need to be reported in both training and test set to see how balance they are.
14. Compare your DNN model with models using logistic regression/supervised learning approach algorithms on your data to show the advantages.
15. To help the clinician visually interpret how the predictive score was generated by the deep-learning model, a monogram is recommended because DNN model is too complicated to interpret in practical settings.
16. The simplified models use a cut-point for each top predictor. The underlying rational of these cut-points needs to be discussed in the discussion section.

Additional comments

Xiaoran Li and colleagues conducted a retrospective study to develop a deep learning model and a risk score model to predict ICU admission and mortality in 1108 hospitalized COVID-19 positive patients using their clinical information. The models were built in 90% of data set and tested in the 10% left. They found that top 5 ICU predictors included procalcitonin, lactate dehydrogenase, C-reactive protein, ferritin, and SpO2, and top 6 mortality predictors included age, lactate dehydrogenase, procalcitonin, cardiac troponin, C-reactive protein, and SpO2. AUCs of the deep- learning model in test dataset were 0.780 for ICU admission and 0.882 for mortality. The simplified risk score model had slightly lower performance, yielding the AUC of 0.728 and 0.848, respectively. The findings of key predictors are not new as they have been reported elsewhere. The models need to be validated in external cohort before practical use. However, the models contribute to the growing need of quantitative tools to stratify hospitalized COVID-19 patients having need of ICU admission and high risk of mortality based on their clinical information.

Although the analysis is interesting and meaningful at the population level, there are still some extra work need to be done.

---

## Round 0.2 · accepted · Accept

Could you attend the comment of Reviewer 3.

Reviewer 1 ·

Basic reporting

The authors have well addressed my concerns and now the paper is ready for publication.

Experimental design

The authors have well addressed my concerns and now the paper is ready for publication.

Validity of the findings

The authors have well addressed my concerns and now the paper is ready for publication.

Additional comments

The authors have well addressed my concerns and now the paper is ready for publication.

Reviewer 2 ·

Basic reporting

The authors have presented satisfactory responses to my queries, and I believe necessary changes have been incorporated to make the paper more informative and clearer now. No more changes are required in my opinion.

Experimental design

No comment

Validity of the findings

No comment

Additional comments

The paper has been substantially improved through the revision, and I believe it is now fit to be accepted in this journal.

Reviewer 3 ·

Basic reporting

The paper is well-written and all relevant figures, tables, and raw data are now provided in desirable presentation.

Experimental design

No additional comment.

Validity of the findings

All underlying data have been provided. The results are well discussed, including the limitation of sample size and the need of external validation.

Additional comments

All my questions have been well addressed. The methods and results are now clearly described.

Minor comment: authors mentioned in the discussion that "Disadvantages of deep learning methods are that it requires comparatively large sample size", and that authors' study has "small sample sizes". So what is the rational when authors decided to choose DNN for developing a predictive model rather than other algorithms that are more suitable for small sample size study, such as Bayesian neural network?